# Equivariant Flow Matching for Molecular Conformer Generation

**Majdi Hassan** [* 1 2]  **Nikhil Shenoy** [* 3 4]  **Jungyoon Lee** [* 1 2]  **Hannes Stärk** [5]  **Stephan Thaler** [4]  **Dominique Beaini** [1 2 4]

## Abstract

Predicting low-energy molecular conformations given a molecular graph is an important but challenging task in computational drug discovery. Existing state-of-the-art approaches either resort to large scale transformer-based models that diffuse over conformer fields, or use computationally expensive methods to generate initial structures and diffuse over torsion angles. In this work, we introduce **E**quivariant **T**ransformer **Flow** (ET-Flow). We showcase that a well-designed flow matching approach with equivariance and harmonic prior alleviates the need for complex internal geometry calculations and large architectures, contrary to the prevailing methods in the field. Our approach results in a straightforward and scalable method that directly operates on all-atom coordinates with minimal assumptions. ET-Flow outperforms or matches the previous state-of-the-art in molecular conformer generation benchmarks with significantly fewer parameters, no dependence on internal geometry, and fast inference.

## 1. Introduction

Generating low-energy 3D representations of molecules, called *conformers*, from the molecular graph is a fundamental task in computational chemistry as the 3D structure of a molecule is responsible for several biological, chemical and physical properties (Guimaraes et al., 2012; Schütt et al., 2018; 2021; Gasteiger et al., 2020; Axelrod & Gomez-Bombarelli, 2023). Conventional approaches to molecular conformer generation consist of stochastic and systematic methods. While stochastic methods such as Molecular Dynamics (MD) accurately generate conformations, they can be slow, cost-intensive, and have low sample diversity (Shim & MacKerell Jr, 2011; Ballard et al., 2015; De Vivo et al.,

---

[*]Equal contribution  [1]Mila - Quebec AI Institute, Montréal, Canada [2]Université de Montréal, Montréal, Canada [3]University of British-Columbia, Vancouver, Canada [4]Valence Labs [5]Massachusets Institute of Technology, Massachusets, USA. Correspondence to: Majdi Hassan <majdi.hassan@umontreal.ca>.

*Accepted at the 1st Machine Learning for Life and Material Sciences Workshop at ICML 2024.* Copyright 2024 by the author(s).

2016; Hawkins, 2017; Pracht et al., 2020). Systematic (rule-based) methods (Hawkins et al., 2010; Bolton et al., 2011; Li et al., 2007; Miteva et al., 2010; Cole et al., 2018; Lagorce et al., 2009) that rely on torsional profiles and knowledge base of fragments are much faster but become less accurate with larger molecules. Therefore, there has been an increasing interest in developing scalable and accurate generative modeling methods in molecular conformer generation.

Existing machine learning based approaches use diffusion models (Ho et al., 2020; Song & Ermon, 2019) to sample diverse and high quality samples given access to low-energy conformations. Prior methods typically fall into two categories: diffusing over atomic coordinates in the Cartesian space (Xu et al., 2022; Wang et al., 2024) or diffusing over internal geometry such as pairwise distances, bond angles, and torsion angles (Ganea et al., 2021; Jing et al., 2022).

Early approaches based on diffusion (Shi et al., 2021; Luo et al., 2021; Xu et al., 2022) faced challenges such as lengthy inference and training times as well as having lower accuracy compared to cheminformatics methods. Torsional Diffusion (Jing et al., 2022) was the first to outperform cheminformatics methods by diffusing only on torsion angles after producing an initial conformer with the chemoinformatics tool RDKiT. This reliance on RDKiT structures instead of employing an end-to-end approach comes with several limitations, such as restricting the tool to applications where the local structures produced by RDKiT are of sufficient accuracy. Unlike prior approaches, the current state-of-the-art MCF (Wang et al., 2024) proposes a domain-agnostic approach by learning to diffuse over functions by scaling transformers and learning soft inductive bias from the data (Zhuang et al., 2022). Consequently, it comes with drawbacks such as high computational demands due to large number of parameters, limited sample efficiency from a lack of inductive biases like euclidean symmetries, and potential difficulties in scenarios with sparse data — a common challenge in this field.

In this paper, we propose **E**quivariant **T**ransformer **Flow** (ET-Flow), a simple yet powerful flow-matching model designed to generate low-energy 3D structures of small molecules with minimal assumptions. We utilize flow matching (Lipman et al., 2022; Albergo et al., 2023; Liu et al., 2022), which enables the learning of arbitrary proba-

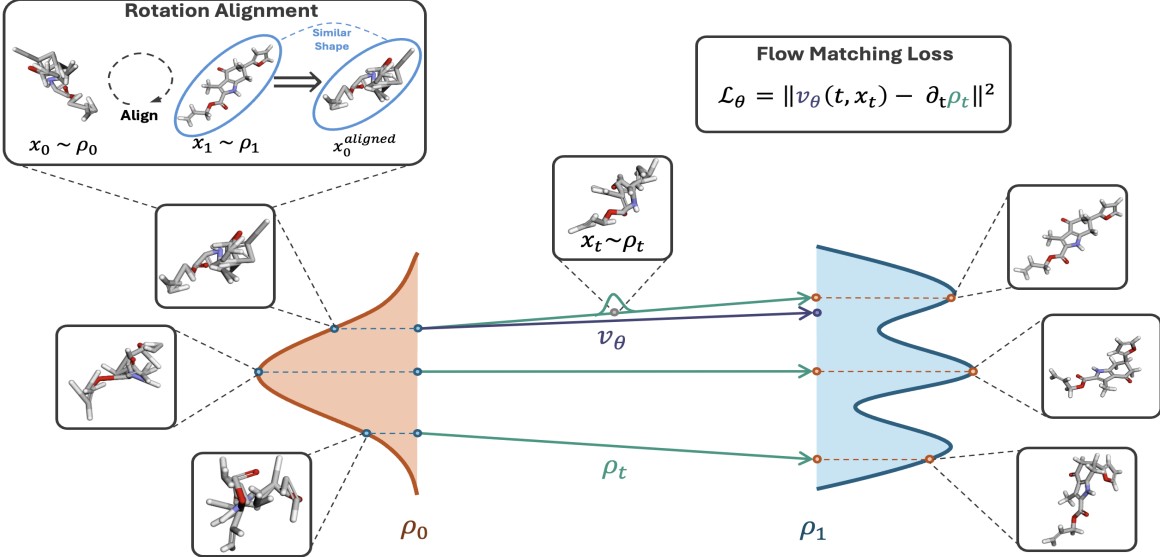

*Figure 1.* Overview of ET-Flow. The model takes as input the features related to the molecular structure like atom and bond features as well as a radius graph of 10Å from the atom positions. Samples are drawn from the harmonic prior and then rotationally aligned with the samples from data. A probability path is constructed between pairs of $x_0$ and $x_1$, and $x_t$ is then sampled from this path at a random time $t$. The network predicts the vector field $\partial_t \rho_t$ given $x_t$ as input while minimizing the flow matching loss.

bility paths beyond diffusion paths, enhancing both training and inference efficiency compared to conventional diffusion generative models. Departing from traditional equivariant architectures like EGNN (Satorras et al., 2021), we adopt an Equivariant Transformer (Thölke & De Fabritiis, 2022) to better capture geometric features. Additionally, our method integrates a Harmonic Prior (Jing et al., 2023; Stark et al., 2023), leveraging the inductive bias that atoms connected by a bond should be in close proximity. We further optimize our flow matching objective by initially conducting rotational alignment on the harmonic prior, thereby constructing shorter probability paths between source and target distributions at minimal computational cost.

Our contributions can be summarized as follows:

1. We obtain state-of-the-art precision results for molecule conformer prediction and improve upon the state-of-the-art by a large margin on ensemble property prediction.

2. Counter to the prevailing status quo in the literature, we provide strong evidence that previous innovations that improved performance at the cost of modeling complexity are not necessary. The significantly simpler but well-engineered ET-Flow can provide better performance.

3. ET-Flow uses orders of magnitude fewer sampling steps than GeoDiff (Xu et al., 2022) and significantly fewer parameters than MCF (Wang et al., 2024).

## 2. Method

We design a scalable equivariant model that generates energy-minimized conformers given a molecular graph. In this section, we layout the framework to achieve this objective by detailing the generative process in flow matching, the rotation alignment between distributions, stochastic sampling, and finally the architecture details.

**Preliminaries** We define notation that we use throughout this paper. Inputs are continuous atom positions $\mathbf{x} \in \mathbb{R}^{N \times 3}$ where $N$ is the number of atoms. We use the notation $v_t(\mathbf{x})$ interchangeably with $v(t, \mathbf{x})$ for vector field.

### 2.1. Flow Matching

The aim is to learn a time-dependent vector field $v_t(x)$ : $\mathbb{R}^{N \times 3} \times [0, 1] \rightarrow \mathbb{R}^{N \times 3}$ associated with the transport map $X_t : \mathbb{R}^{N \times 3} \times [0, 1] \rightarrow \mathbb{R}^{N \times 3}$ that pushes forward samples from a base distribution $\rho_0$, often an easy-to-sample distribution, to samples from a more complex target distribution $\rho_1$, the low-energy conformations of a molecule. This can be defined as an ordinary differential equation (ODE),

$$\dot{X}_t(\mathbf{x}) = v_t(X_t(\mathbf{x})), \qquad X_{t=0} = \mathbf{x}_0, \qquad (1)$$

where $x_0 \sim \rho_0$. We can construct the $v_t$ via a time-differentiable interpolation between samples from $\rho_0$ and $\rho_1$ that gives rise to a probability path $\rho_t$ that we can easily sample (Lipman et al., 2022; Liu et al., 2022; Albergo & Vanden-Eijnden, 2023; Tong et al., 2023). The general interpolation between samples $x_0 \sim \rho_0$ and $x_1 \sim \rho_1$ can be

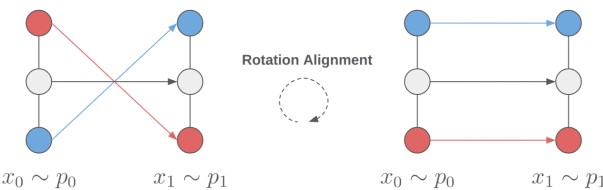

*Figure 2.* Rotation alignment using the Kabsh algorithm (Kabsch, 1976) resulting in shorter and straighter paths between the base and target structures.

defined as:

$$I_t(\mathbf{x}_0, \mathbf{x}_1) = \alpha_t \mathbf{x}_1 + \beta_t \mathbf{x}_0. \tag{2}$$

Given this interpolant, we can define the probability path $\rho_t(x) = \mathcal{N}(x|I_t(\mathbf{x}_0, \mathbf{x}_1), \sigma_t^2 \mathbf{I})$, and the vector field can be computed as $v_t(\mathbf{x}) = \partial_t \rho_t(\mathbf{x})$ which has the following form

$$v_t(\mathbf{x}) = \dot{\alpha}_t \mathbf{x}_1 + \dot{\beta}_t \mathbf{x}_0 + \dot{\sigma}_t \mathbf{z} \qquad \mathbf{z} \sim \mathcal{N}(0, \mathbf{I}). \tag{3}$$

In our work, we use linear interpolation where $\alpha_t = t$, $\beta_t = 1 - t$, and $\sigma_t = \sigma\sqrt{t(1-t)}$, resulting in the vector field

$$v_t(\mathbf{x}) = \mathbf{x}_1 - \mathbf{x}_0 + \frac{1 - 2t}{2\sqrt{t(1-t)}}\mathbf{z}. \tag{4}$$

Now, we can define the objective function for learning a vector field $v_\theta(\mathbf{x})$ that generates a probability path $\rho_t$ between a base density $\rho_0$ and the target density $\rho_1$ as,

$$\mathcal{L} = \mathbb{E}_{t \sim \mathcal{U}(0,1), \mathbf{x} \sim \rho_t(\mathbf{x}_0, \mathbf{x}_1)} \|v(t, \mathbf{x}) - v_\theta(t, \mathbf{x})\|^2. \tag{5}$$

For training, we sample (i) $\mathbf{x}_0 \sim \rho_0$, $\mathbf{x}_1 \sim \rho_1$, and $t \sim \mathcal{U}(0,1)$, (ii) interpolate according to Equation 2, (iii) add noise from a standard Gaussian, and (iv) minimize the loss defined in Equation 5. For sampling, we sample $\mathbf{x}_0 \sim \rho_0$ and integrate from $t = 0$ to $t = 1$ using the Euler's method. At each time-step, the Euler solver iteratively predicts the vector field for $\mathbf{x}_t$ and updates its position $\mathbf{x}_{t+\Delta t} = \mathbf{x}_t + v_\theta(t, \mathbf{x})\Delta t$. More details on the training and sampling algorithms are provided in Appendix D.

### 2.2. Alignment

Several previous works (Tong et al., 2023; Klein et al., 2024; Jing et al., 2024; Song et al., 2024) demonstrate that constructing a straighter path between base distribution $\rho_0$ and target distribution $\rho_1$ minimizes the transport costs and improves performance. In our work, we reduce the transport costs between samples from the harmonic prior $\rho_0$ and samples from the data distribution $\rho_1$ by rotationally aligning them using the Kabsch algorithm (Kabsch, 1976) similar to (Klein et al., 2024; Jing et al., 2024). This approach leads to faster convergence and reduces the path length between

atoms by leveraging the similarity in "shape" of the samples as seen in Figure 1 and Figure 2 without incurring high computational cost.

### 2.3. Stochastic Sampling

We employ a variant of the stochastic sampling technique inspired by (Karras et al., 2022). Specifically, we inject noise at each time step, evaluate the vector field from the intermediate state after adding noise, and then perform the deterministic ODE step. The original method utilizes a second-order integration, which averages the denoiser output at the noisy intermediate state and the state at the next time step after integration. In our experiment, we use the stochastic sampler without this second-order correction term, which empirically provided a performance boost comparable to the second-order method. We apply stochastic sampling only during the final part of the integration steps, specifically within the range $t \in [0.8, 1.0]$. This helps prevent drifting towards overpopulated density regions and improves the quality of the samples, as noted by (Karras et al., 2022). Stochastic sampling has improved both diversity and accuracy of the generated conformers, measured by Coverage and Average Minimum RMSD (AMR) respectively as shown in Table 1. Detailed information on the stochastic sampling algorithm is provided in Appendix D.

### 2.4. Chirality Correction

While generating conformations, it is necessary to take account of the stereochemistry of atoms bonded to four distinct groups also referred to as tetrahedral chiral centers. Molecules that only differ in the orientation of these chiral centers are enantiomers and are mirror images of each other. The orientated volume (OV) of a chirality center can be computed based on the ordered coordinates of the distinct groups (say $\boldsymbol{p}_1, \boldsymbol{p}_2, \boldsymbol{p}_3, \boldsymbol{p}_4$) as,

$$OV(\boldsymbol{p}_1, \boldsymbol{p}_2, \boldsymbol{p}_3, \boldsymbol{p}_4) = sign\left(\begin{vmatrix} 1 & 1 & 1 & 1 \\ x_1 & x_2 & x_3 & x_4 \\ y_1 & y_2 & y_3 & y_4 \\ z_1 & z_2 & z_3 & z_4 \end{vmatrix}\right). \tag{6}$$

We also have access to the required orientation for each chiral center using the chirality tag provided by RDKit. In our work, once we generate conformations, we do a *post hoc* correction where we compute orientation volume for each chiral center and compare it with the chiral tag provided by RDKit. If there is a mismatch, we simply flip the conformation against the z-axis. This is similar to the orientation correction done for generated local structure in GeoMol (Ganea et al., 2021).

*Table 1.* Molecule conformer generation results on GEOM-DRUGS ($\delta = 0.75$Å). SS is ET-Flow with stochastic sampling. For both ET-Flow and ET-Flow-SS, we sample conformations over 50 time-steps.

| | Recall | | | | Precision | | | |
|---|---|---|---|---|---|---|---|---|
| | Coverage ↑ | | AMR ↓ | | Coverage ↑ | | AMR ↓ | |
| | mean | median | mean | median | mean | median | mean | median |
| GeoDiff | 42.10 | 37.80 | 0.835 | 0.809 | 24.90 | 14.50 | 1.136 | 1.090 |
| GeoMol | 44.60 | 41.40 | 0.875 | 0.834 | 43.00 | 36.40 | 0.928 | 0.841 |
| Torsional Diff. | 72.70 | 80.00 | 0.582 | 0.565 | 55.20 | 56.90 | 0.778 | 0.729 |
| MCF - S (13M) | 79.4 | 87.5 | 0.512 | 0.492 | 57.4 | 57.6 | 0.761 | 0.715 |
| MCF - B (62M) | 84.0 | 91.5 | 0.427 | 0.402 | 64.0 | 66.2 | 0.667 | 0.605 |
| MCF - L (242M) | **84.7** | **92.2** | **0.390** | **0.247** | 66.8 | 71.3 | 0.618 | 0.530 |
| ET-Flow (8.3M) | 79.53 | 84.57 | 0.452 | 0.419 | 74.38 | 81.04 | 0.541 | 0.470 |
| ET-Flow - SS (8.3M) | 79.62 | 84.63 | 0.439 | 0.406 | **75.19** | **81.66** | **0.517** | **0.442** |

*Table 2.* Ablation over number of inference steps on GEOM-DRUGS ($\delta = 0.75$Å). Performance of ET-Flow at 5 steps is competent across all metrics while also retaining state-of-the-art performance on precision metrics when compared with previous methods.

| | Recall | | | | Precision | | | |
|---|---|---|---|---|---|---|---|---|
| | Coverage ↑ | | AMR ↓ | | Coverage ↑ | | AMR ↓ | |
| | mean | median | mean | median | mean | median | mean | median |
| ET-Flow (5 Steps) | 77.84 | 82.21 | 0.476 | 0.443 | 74.03 | 80.8 | 0.55 | 0.474 |
| ET-Flow (10 Steps) | 79.05 | 84.00 | 0.451 | 0.415 | 74.64 | **81.38** | 0.533 | 0.457 |
| ET-Flow (20 Steps) | 79.29 | 84.04 | **0.449** | **0.413** | **74.89** | 81.32 | **0.531** | **0.454** |
| ET-Flow (50 Steps) | **79.53** | **84.57** | 0.452 | 0.419 | 74.38 | 81.04 | 0.541 | 0.470 |

## 2.5. Architecture

We utilize the equivariant transformer architecture as proposed in the TorchMD-NET (Thölke & De Fabritiis, 2022) which is designed using similar principles as the original Transformer (Vaswani et al., 2017) architecture. The architecture comprises of 3 blocks. First, an embedding layer encodes the inputs (atomic positions, atomic numbers, atom features, edge features, the edge connections and time-step) into a set of invariant features. Initial equivariant features are constructed using normalized edge vectors. Second, a series of update layers update both the invariant and equivariant features using a multi-head attention mechanism. Finally, the Output Layer outputs the vector field by updating the equivariant features using gated equivariant blocks (Schütt et al., 2018). We provide additional details on the model implementation in Appendix B.

## 3. Experiments

We empirically evaluate ET-Flow by comparing the generated and ground-truth conformers in terms of distance-based RMSD (Section 3.2) and chemical property based metrics (Section 3.3). We present the general experimental setups in Section 3.1, and we discuss the importance of our design

choices by ablation studies in Section C.1. The implementation details are provided in Appendix B.

### 3.1. Experimental Setup

**Dataset**: We conduct our experiments on the GEOM dataset (Axelrod & Gomez-Bombarelli, 2022), which offers curated conformer ensembles produced through meta-dynamics in CREST (Pracht et al., 2024). Our primary focus is on GEOM-DRUGS, the most extensive and pharmacologically relevant subset comprising 304k drug-like molecules, each with an average of 44 atoms. We use a train/validation/test (243473/30433/1000) split as provided in (Ganea et al., 2021) Additionally, we train and test model on GEOM-QM9, a subset of smaller molecules with an average of 11 atoms. Finally, in order to assess the model's ability to generalize to larger molecules, we evaluate the model trained on DRUGS on a GEOM-XL dataset, a subset of large molecules with more than 100 atoms. The results for GEOM-QM9 and GEOM-XL can be found in Appendix C.

**Evaluation**: Our evaluation methodology is similar to that of (Jing et al., 2022). First, we look at RMSD based metrics like Coverage and Average Minimum RMSD (AMR) between generated and ground truth conformer ensembles. For this, we generate $2K$ conformers for a molecule with $K$

ground truth conformers. Second, we look at chemical similarity using properties like Energy ($E$), dipole moment ($\mu$), HOMO-LUMO gap ($\Delta\epsilon$) and the minimum energy ($E_{\min}$) calculated using xTB (Bannwarth et al., 2019).

**Baselines**: We benchmark ET-Flow against leading approaches outlined in Section A. Specifically, we assess the performance of GeoMol (Ganea et al., 2021), GeoDiff (Xu et al., 2022), Torsional Diff (Jing et al., 2022), and MCF (Wang et al., 2024). Notably, the most recent among these, MCF, has demonstrated superior performance across evaluation metrics compared to its predecessors. It is worth mentioning that GeoDiff initially utilized a limited subset of the DRUGS dataset; thus, for a fair comparison, we consider its re-evaluated performance as presented in (Jing et al., 2022).

### 3.2. Ensemble RMSD

As shown in Table 1 and Section C.2, ET-Flow outperforms all preceding methods and demonstrates competitive performance with the previous state-of-the-art, MCF (Wang et al., 2024). Despite being significantly smaller with only 8.3M parameters, ET-Flow shows a substantial improvement in the quality of generated conformers, as evidenced by superior Precision metrics across all MCF models, including the largest MCF-L. When compared to MCF-S, which is closer in size, ET-Flow achieves better Precision while the impact on Recall is less significant and limited to Recall Coverage. Notably, our Recall AMR remains competitive with much bigger MCF-B, underscoring the inherent advantage of our method in accurately predicting overall structures.

### 3.3. Ensemble Properties

RMSD provides a geometric measure for assessing ensemble quality, but it is also essential to consider the chemical similarity between generated and ground truth ensembles. For a random 100-molecule subset of DRUGS, if a molecule has $K$ ground truth conformers, we generate a minimum of $2K$ and a maximum of 32 conformers per molecule. These conformers are then relaxed using GFN2-xTB (Bannwarth et al., 2019), and the Boltzmann-weighted properties of the generated and ground truth ensembles are compared. Specifically, using xTB (Bannwarth et al., 2019), we compute properties such as energy ($E$), dipole moment ($\mu$), HOMO-LUMO gap ($\Delta\epsilon$), and the minimum energy ($E_{min}$). Table 3 illustrates the median errors for ET-Flow and the baselines, highlighting our method's capability to produce chemically accurate ensembles. Notably, we achieve significant improvements over both TorsionDiff and MCF across all evaluated properties.

*Table 3.* Median averaged errors of ensemble properties between sampled and generated conformers ($E$, $\Delta\varepsilon$, $E_{min}$ in kcal/mol, and $\mu$ in debye).

|  | $E$ | $\mu$ | $\Delta\epsilon$ | $E_{\min}$ |
|---|---|---|---|---|
| OMEGA | 0.68 | 0.66 | 0.68 | 0.69 |
| GeoDiff | 0.31 | 0.35 | 0.89 | 0.39 |
| GeoMol | 0.42 | 0.34 | 0.59 | 0.40 |
| Torsional Diff. | 0.22 | 0.35 | 0.54 | 0.13 |
| MCF | 0.68±0.06 | 0.28±0.05 | 0.63±0.05 | 0.04±0.00 |
| ET-Flow | **0.18±0.01** | **0.18±0.01** | **0.35±0.06** | **0.02±0.00** |

### 3.4. Inference Steps Ablation

In Table 1, our sampling process with ET-Flow utilizes 50 inference steps. To evaluate the method's performance under constrained computational resources, we conducted an ablation study by progressively reducing the number of inference steps. Specifically, we sample for 5, 10 and 20 time-steps. The results on GEOM-DRUGS are presented in Table 2, showing minimal performance degradation with fewer sampling steps. Notably, ET-Flow demonstrates high efficiency, maintaining performance across all precision and recall metrics even with as few as 5 inference steps. Interestingly, ET-Flow with 5 steps still achieves superior precision metrics compared to all existing methods. This underscores ET-Flow's ability to generate high-quality conformations while operating within limited computational budgets.

## 4. Conclusion

In this paper, we present ET-Flow, a simple and scalable method, which utilizes an equivariant transformer with flow matching to achieve state-of-the-art performance on molecular conformer generation benchmarks. We emphasize that incorporating inductive biases, such as equivariance, and enhancing probability paths with a harmonic prior and RMSD alignment, enables us to attain these results while maintaining parameter and sample efficiency.

**Future Works**: While ET-Flow demonstrates competitive performance in molecular conformer generation, there are areas where it can be improved. We propose three future directions. First, we observe that a well-designed sampling process incorporating stochasticity can enhance the quality and diversity of generated samples. An extension of our method could involve using Stochastic Differential Equations (SDEs), which utilize both vector field and score in inference, potentially improving the diversity of samples (Ma et al., 2024). Second, we propose to scale the number of parameters of ET-Flow, which has been shown to be useful in molecular conformer generation for MCF (Wang et al., 2024), especially for out-of-distribution generation in GEOM-XL. Lastly, we aim to alleviate the need for an additional chirality correction step via architectural changes.

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

# A. Related Works

**Diffusion Generative Models** Diffusion models (Song & Ermon, 2019; Song et al., 2020; Ho et al., 2020) enables a high-quality and diverse sampling from an unknown data distribution by approximating the Stochastic Differential Equation (SDE) that maps a simple density i.e. Gaussian to the unknown data density. Concretely, it involves training a neural network to learn the score, represented as $\nabla_{\mathbf{x}} \log p_t(\mathbf{x})$ of the diffused data. During inference, the model generates sample by iteratively solving the reverse SDE. However, diffusion models have inherent drawbacks, as they (i) require on longer training times (ii) are restricted to specific probability paths and (iii) depend on the use of complicated tricks to speed up sampling (Song et al., 2020; Zhang & Chen, 2022).

**Flow Matching** Continuous normalizing flows(CNFs) is a generative method that is capable of modeling arbitrary probability path, including the probability paths modeled by diffusion processes. Flow Matching (Albergo et al., 2023; Lipman et al., 2022; Liu et al., 2022) provides a general framework to learn CNFs, while improving upon the speed of diffusion training and inference. Through simple regression against the vector field reminiscent of the score-matching objective in diffusion models, Flow matching has enabled a fast, simulation-free training of CNFs. Several subsequent studies have then expanded the scope of flow matching objective to manifolds (Chen & Lipman, 2024), arbitrary sources (Pooladian et al., 2023), and conditional flow matching with arbitrary transport maps and optimal couplings between source and target samples (Tong et al., 2023).

**Molecular Conformer Generation** Various machine learning (ML) based approaches (Kingma & Welling, 2013; Liberti et al., 2014; Dinh et al., 2016; Simm & Hernández-Lobato, 2019; Shi et al., 2021; Luo et al., 2021; Xu et al., 2021; Ganea et al., 2021; Xu et al., 2022; Jing et al., 2022; Wang et al., 2023) have been developed to improve upon the limitations of conventional methods, among which the most advanced are TorsionDiff (Jing et al., 2022) and Molecular Conformer Fields (MCF) (Wang et al., 2024). TorsionDiff designs a diffusion model on the torsion angles while incorporating the local structure from RDKiT ETKDG (Riniker & Landrum, 2015). MCF trains a diffusion model over functions, specifically functions that map elements from the molecular graph to points in 3D space.

**Equivariant Architectures for Atomistic Systems** Inductive biases play an important role in generalization and sample efficiency. In the case of 3D atomistic modelling, one example of a useful inductive bias is the euclidean group $SO(3)$ which represents rotation equivariance in 3D space. Recently, various equivariant architectures (Duval et al., 2023) have been developed that act on both Cartesian (Satorras et al., 2021; Thölke & De Fabritiis, 2022; Simeon & De Fabritiis, 2024; Du et al., 2022; Frank et al., 2022) and spherical basis (Musaelian et al., 2023; Batatia et al., 2022; Fuchs et al., 2020; Liao et al., 2023; Passaro & Zitnick, 2023; Anderson et al., 2019; Thomas et al., 2018). For molecular conformer generation, initial methods like ConfGF, DGSM utilize invariant networks as they act upon inter-atomic distances, whereas the use of equivariant GNNs have been used in GeoDiff (Xu et al., 2022) and Torsional Diffusion (Jing et al., 2022). GeoDiff utilizes EGNN (Satorras et al., 2021), a Cartesian basis equivariant architecture while Torsional Diffusion uses Tensor Field Networks (Thomas et al., 2018) to output pseudoscalars.

# B. Implementation Details

## B.1. Architecture

The ET-Flow architecture (Figure 3) consists of 3 major components, an embedding layer, update layers and an output layer. We use a modified version of the embedding and update layers from the equivariant transformer architecture of TorchMD-NET (Thölke & De Fabritiis, 2022) whereas the output layer utilizes the gated equivariant blocks from (Schütt et al., 2018). We highlight our modifications over the original architectures with this color. These modifications enable stabilized training since we use a larger network than the one proposed in the TorchMD-NET (Thölke & De Fabritiis, 2022) paper. Additionally, since our input structures are interpolations between structures sampled from a prior and actual conformations, it is important to ensure our network is numerically stable when the interpolations contain two atoms very close to each other.

**Embedding Layer**: This layer learns an embedding for the $i$-th atom using the atomic number ($z_i$), atomic attributes ($h_i$), time-step $t$, edge features $l_{ij}$ and neighborhood embedding $n_i$. Atomic numbers are embedded into two vectors, one for atomic number embedding and the other for neighboorhood embedding $n_i$. The atomic attributes $h_i$ are projected into a feature vector using a simple 2-layer Multi-Layer Perceptron (MLP). The neighboorhood embedding for an atom $n_i$ is

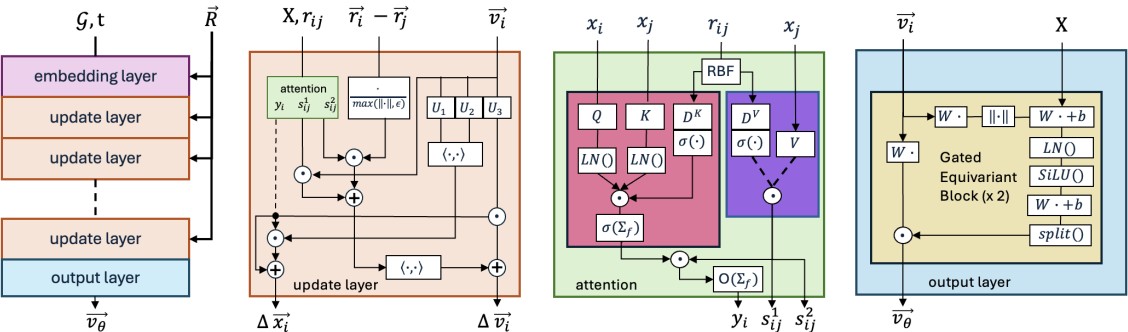

*Figure 3.* (a) Overall Architecture of ET-Flow consisting of 3 components, (1) Embedding Layer (2) Update layers (with attention layers) and (3) Output Layer. (b) Update Layer with all the operations involved, (c) Attention block modified with LayerNorm and (d) Output Layer consisting of Gated Equivariant Blocks from Schütt et al. (2018)

computed as,

$$z_i = \text{embed}^{\text{int}}(z_i) \tag{7}$$

$$h_i = \text{MLP}(h_i) \tag{8}$$

$$n_i = \sum_{j=1}^{N} \text{embed}^{\text{nbh}}(z_j) \cdot \text{g}(d_{ij}, l_{ij}) \tag{9}$$

where the second term $g(d_{ij}, l_{ij})$ is a edge based feature, which is a linear projection of a vector concatenating $k$ exponential radial basis functions multiplied with a cutoff function $(\phi_{d_{ij}})$ and edge features $l_{ij}$. It is computed as,

$$g(d_{ij}, l_{ij}) = W^F \left[ \phi(d_{ij}) e_1^{\text{RBF}}(d_{ij}) .... \phi(d_{ij}) e_K^{\text{RBF}}(d_{ij}), l_{ij} \right] \tag{10}$$

$$\phi(d_{ij}) = \begin{cases} \frac{1}{2} \left( \cos(\frac{\pi d_{ij}}{d_{cutoff}} + 1) \right), & \text{if } d_{ij} \leq d_{\text{cutoff}} \\ 0, & \text{otherwise} \end{cases} \tag{11}$$

The hyperparameters for the radial basis functions are set as proposed in (Unke & Meuwly, 2019). Once the neighboorhood embedding $n_i$ is computed, the final atom-level embedding is computed via a linear projection as,

$$x_i = W^C \left[ \text{embed}^{\text{int}}(z_i), h_i, t, n_i \right] \tag{12}$$

**Attention Mechanism**: The multi-head dot-product attention operation uses atom features $x_i$, atom attributes $h_i$, time-step $t$ and inter-atomic distances $r_{ij}$ to compute attention weights. The input atom-level features $x_i$ are mixed with the atom attributes $h_i$ and the time-step $t$ using an MLP and then further normalized using a LayerNorm (Ba et al., 2016). To compute the attention matrix, the inter-atomic distances $r_{ij}$ are projected into two dimensional filters $D^K$ and $D^V$ as,

$$D^K = \sigma \left( W^{D^K} e^{RBF}(r_{ij}) + b^{D^K} \right)$$

$$D^V = \sigma \left( W^{D^V} e^{RBF}(r_{ij}) + b^{D^V} \right) \tag{13}$$

The atom level features are then linearly projected along with a LayerNorm operation to derive the query $Q$ and key $K$ vectors. The value vector $V$ is computed with only the linear projection of atom-level features. Applying LayerNorm on Q, K vectors (also referred to as QK-Norm) has proven to stabilize un-normalized values in the attention matrix (Dehghani et al., 2023; Esser et al., 2024) when scaling networks to large number of parameters. The $Q$ and $K$ vectors are then used along with the distance filter $D^K$ for a dot-product operation over the feature dimension,

$$Q = \text{LayerNorm}(W^Q x_i), \quad K = \text{LayerNorm}(W^K x_i), \quad V = W^V x_i \tag{14}$$

$$\text{dot}(Q, K, D^K) = \sum_k^F Q_k \cdot K_k \cdot D_k^K. \tag{15}$$

The attention matrix is derived by passing the above dot-product operation matrix through a non-linearity and weighting it using a cosine cutoff $\phi(d_{ij})$ (similar to Equation 11) which ensure the attention weights are non-zero only when two atoms are within a specified cutoff,

$$A = \text{SiLU}(\text{dot}(Q, K, D^K)) \cdot \phi(d_{ij}). \tag{16}$$

Using the value vector $V$ and the distance filter $D_V$, we derive 3 equally sized filters by splitting along the feature dimension $(s_{ij}^1, s_{ij}^2, s_{ij}^3 = \text{split}(V_j \cdot D_{ij}^V))$. A linear projection is then applied on to combine the attention matrix and the vectors $s_{ij}^3$ to derive a atom level level feature $y_i$ $(= W^O \left(\sum_j^N A_{ij} \cdot s_{ij}^3\right))$. The output of the attention operation are $y_i$ (an atom level feature) and two scalar filters $s_{ij}^1$ and $s_{ij}^2$ (edge-level features).

**Update Layer**: The update layer computes interactions between atoms in the atttention block and uses the outputs to update the scalar feature $x_i$ and the vector feature $\vec{v}_i$. First, the scalar feature output $y_i$ from the attention mechanism is split into 3 features $(q_i^1, q_i^2, q_i^3)$, out of which $q_i^1$ and $q_i^2$ are used for the scalar feature update as,

$$\Delta x_i = q_i^1 + q_i^2 \cdot \langle U_1 \vec{v}_i \cdot U_2 \vec{v}_i \rangle, \tag{17}$$

where $\langle U_1 \vec{v}_i \cdot U_2 \vec{v}_i \rangle$ is the inner product between linear projections of vector features $\vec{v}_i$ with matrices $U_1, U_2$.

The edge vector update consists of two components. First, we compute a vector $\vec{w}_i$, which for each atom is computed as a weighted sum of vector features and a clamped-norm of the edge vectors over all neighbors. Finally, a sum over $\vec{w}_i$ and a linear projection of the vector feature gives us,

$$\vec{w}_i = \sum_j^N s_{ij}^1 \cdot \vec{v}_j + s_{ij}^2 \cdot \frac{\vec{r}_i - \vec{r}_j}{\max(\|\vec{r}_i - \vec{r}_j\|, \epsilon)}, \tag{18}$$

$$\Delta \vec{v}_i = \vec{w}_i + q_i^3 \cdot U_3 \vec{v}_i \tag{19}$$

where $U_1$ and $U_3$ are projection vectors over the feature dimension of the vector feature $\vec{v}_i$. Our main modifications in this layer, we clamp the minimum value of the norm (to $\epsilon = 0.01$) to prevent numerically large values in cases where positions of two atoms are sampled two close from the prior.

**Output Layer**: The output layer consists of Gated Equivariant Blocks from (Schütt et al., 2018). Given atom scalar $x_i$ and vector features $\vec{v}_i$, the updates in each block is defined as,

$$x_{i,\text{updated}}, \vec{w}_i = \text{split}(\text{MLP}([x_i, U_1 \vec{v}_i])) \tag{20}$$

$$\vec{v}_{i,\text{updated}} = (U_2 \vec{v}_i) \cdot \vec{w}_i \tag{21}$$

$$\tag{22}$$

Here, $U_1$ and $U_2$ are linear projection matrices that act along feature dimension. Our modification is to use LayerNorm in the MLP to improve training stability.

## B.2. Input Featurization

Atomic features (or Node Features) are computed using RDKit (Landrum et al., 2013) features as described in Table 4. For computing edge features and edge index, we use a combination of global (radius based edges) and local (molecular graph edges) similar to Jing et al. (2022).

## B.3. Evaluation Metrics

Following the approaches of (Ganea et al., 2021; Xu et al., 2022; Jing et al., 2022), we utilize Average Minimum RMSD (AMR) and Coverage (COV) to assess the performance of molecular conformer generation. Here, $C_g$ denotes the set of generated conformations, and $C_r$ denotes the set of reference conformations. For both AMR and COV, we calculate and report Recall (R) and Precision (P). Recall measures the extent to which the generated conformers capture the ground-truth conformers, while Precision indicates the proportion of generated conformers that are accurate. The specific formulations for these metrics are detailed in the following equations:

$$\text{AMR-R}(C_g, C_r) = \frac{1}{|C_r|} \sum_{\mathbf{R} \in C_r} \min_{\hat{\mathbf{R}} \in C_g} \text{RMSD}(\mathbf{R}, \hat{\mathbf{R}})$$

*Table 4.* Atomic features included in ET-Flow.

| Name | Description | Range |
|------|-------------|-------|
| chirality | Chirality Tag | {unspecified, tetrahedral CW, tetrahedral CCW, other} |
| degree | Number of bonded neighbors | $\{x : 0 \le x \le 10, x \in Z\}$ |
| charge | Formal charge of atom | $\{x : -5 \le x \le 5, x \in Z\}$ |
| num_H | Total Number of Hydrogens | $\{x : 0 \le x \le 8, x \in Z\}$ |
| number_radical_e | Number of Radical Electrons | $\{x : 0 \le x \le 4, x \in Z\}$ |
| hybrization | Hybrization type | {sp, sp$^2$, sp$^3$, sp$^3$d, sp$^3$d$^2$, other} |
| aromatic | Whether on a aromatic ring | {True, False} |
| in_ring | Whether in a ring | {True, False} |

$$\text{COV-R}(C_g, C_r) = \frac{1}{|C_r|} |\{\mathbf{R} \in C_r | \text{RMSD}(\mathbf{R}, \hat{\mathbf{R}}) < \delta, \hat{\mathbf{R}} \in C_g\}|$$

$$\text{AMR-P}(C_r, C_g) = \frac{1}{|C_g|} \sum_{\hat{\mathbf{R}} \in C_g} \min_{\mathbf{R} \in C_r} \text{RMSD}(\hat{\mathbf{R}}, \mathbf{R})$$

$$\text{COV-P}(C_r, C_g) = \frac{1}{|C_g|} |\{\hat{\mathbf{R}} \in C_g | \text{RMSD}(\hat{\mathbf{R}}, \mathbf{R}) < \delta, \mathbf{R} \in C_r\}|$$

A lower AMR score signifies improved accuracy, while a higher COV score reflects greater diversity in the generative model. Following (Jing et al., 2022), the threshold $\delta$ is set to $0.5\mathring{A}$ for GEOM-QM9 and $0.75\mathring{A}$ for GEOM-DRUGS.

### B.4. Training Details and Hyperparameters

*Table 5.* Hyperparameters for ET-Flow

| Hyper-parameter | ET-Flow |
|-----------------|---------|
| num_layers | 20 |
| hidden_channels | 160 |
| num_heads | 8 |
| neighbor_embedding | True |
| cutoff_lower | 0.0 |
| cutoff_higher | 10.0 |
| node_attr_dim | 8 |
| edge_attr_dim | 1 |
| reduce_op | True |
| activation | SiLU |
| attn_activation | SiLU |
| # param | 8.3M |

For GEOM-DRUGS, we train ET-Flow for a fixed 250 epochs with a batch size of 64 and 5000 training batches per epoch per GPU on 8 A100 GPUs. For the learning rate, we use the Adam Optimizer with a cosine annealing learning rate which goes from a maximum of $1.e-3$ to a minimum $1.e-7$ over 250 epochs with a weight decay of $1.e-10$. For GEOM-QM9, we train ET-Flow for 200 epochs with a batch size of 128, and use all of the training dataset per epoch on 4 A100 GPUs. We use the cosine annealing learning rate schedule with maximum of $8.e-4$ to minimum of $1.e-7$ over 100 epochs, post which the maximum is reduced by a factor of 0.05.

We select checkpoints based on the lowest validation error. For ablations, experiments are conducted using models trained on 4 A100 GPUs for 50 epochs with a learning rate of $1e-4$ on GEOM-DRUGS. The hyperparameters for the experiments are shared in Table 5.

## C. Additional Results

### C.1. Design Choice Ablations

We conduct a series of ablation studies to assess the influence of each component in the ET-Flow. Particularly, we re-run the experiments with (1) $O(3)$ equivariance without chirality correction, (2) Absence of Alignment, (3) Gaussian Prior as a base distribution. We demonstrate that improving probability paths and utilizing an expressive equivariant architecture with correct symmetries are key components for ET-Flow to achieve state of the art performance. The ablations were ran with reduced settings (50 epochs; lr $= 1e - 4$; 4 A100 gpus). Results are shown in Table C.1.

*Table 6.* Ablation results on GEOM-DRUGS.

|  | Recall | | | | Precision | | | |
|---|---|---|---|---|---|---|---|---|
|  | Coverage ↑ | | AMR ↓ | | Coverage ↑ | | AMR ↓ | |
|  | mean | median | mean | median | mean | median | mean | median |
| ET-Flow | 75.37 | 82.35 | 0.557 | 0.529 | 58.90 | 60.87 | 0.742 | 0.690 |
| ET-Flow ($O(3)$) | 72.74 | 79.21 | 0.576 | 0.556 | 54.84 | 54.11 | 0.794 | 0.739 |
| ET-Flow (w/o Alignment) | 68.67 | 74.71 | 0.622 | 0.611 | 47.09 | 44.25 | 0.870 | 0.832 |
| ET-Flow (Gaussian Prior) | 66.53 | 73.01 | 0.640 | 0.625 | 44.41 | 40.88 | 0.903 | 0.864 |

### C.2. Coverage Threshold Plots

We show a breakdown of the performance on GEOM-DRUGS of ET-Flow against Torsional diffusion (Jing et al., 2022) as a function of the threshold distance in Section C.2. Across a wide range of threshold values, ET-Flow consistently outperforms Torsional Diffusion in terms of both recall and precision, except for slight underperformance at very high threshold values. Particularly notable is ET-Flow's superior performance at lower threshold regions, highlighting its ability for accurate conformer prediction. We were unable to replicate the results of MCF (Wang et al., 2024) as their codebase is currently not publicly available; therefore, we did not include them in our comparison plot.

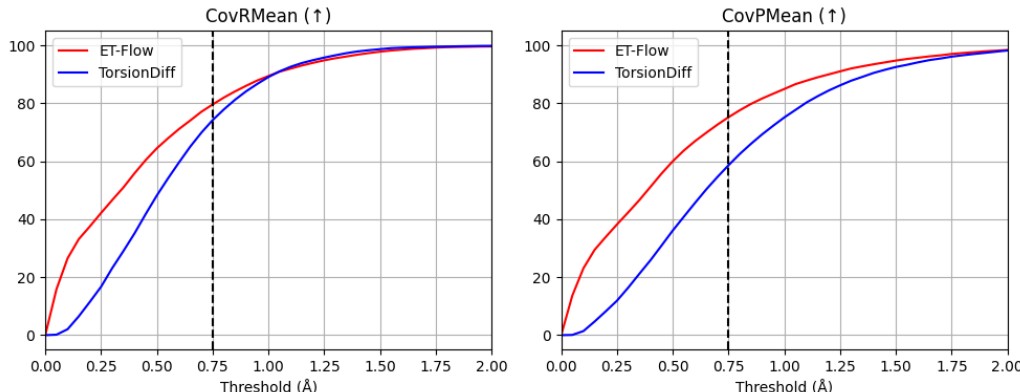

*Figure 4.* Recall and Precision Coverage result on GEOM-DRUGS as a function of the threshold distance. ET-Flow outperforms TorsionDiff by a large margin especially in a lower threshold region.

### C.3. Results on GEOM-QM9

We also train and evaluate our model on the small molecules from GEOM-QM9, with the performance reported in Table 7. ET-Flow consistently outperforms or matches the previous state-of-the-art, MCF, across all metrics, despite its significantly more compact model size.

*Table 7.* Molecule conformer generation results on GEOM-QM9 ($\delta = 0.5$Å).

| | Recall | | | | Precision | | | |
| | Coverage ↑ | | AMR ↓ | | Coverage ↑ | | AMR ↓ | |
| | mean | median | mean | median | mean | median | mean | median |
|---|---|---|---|---|---|---|---|---|
| CGCF | 69.47 | 96.15 | 0.425 | 0.374 | 38.20 | 33.33 | 0.711 | 0.695 |
| GeoDiff | 76.50 | **100.00** | 0.297 | 0.229 | 50.00 | 33.50 | 1.524 | 0.510 |
| GeoMol | 91.50 | **100.00** | 0.225 | 0.193 | 87.60 | **100.00** | 0.270 | 0.241 |
| Torsional Diff. | 92.80 | **100.00** | 0.178 | 0.147 | 92.70 | **100.00** | 0.221 | 0.195 |
| MCF | **95.0** | 100.00 | 0.103 | 0.044 | **93.7** | **100.00** | 0.119 | 0.055 |
| ET-Flow (ours) | 94.99 | **100.00** | **0.083** | **0.035** | 91.00 | **100.00** | **0.116** | **0.047** |

## C.4. Results on GEOM-XL

We now assess how well a model trained on GEOM-DRUGS generalises to unseen molecules with large numbers of atoms, using the GEOM-XL dataset containing a total of 102 molecules. This provides insights into the model's capacity to tackle larger molecules and out-of-distribution tasks. Upon executing the checkpoint provided by Torsional Diffusion, we encountered 27 failed cases for generation likely due to RDKit failures, similar to the observations in MCF albeit with slightly different exact numbers. In both experiments involving all 102 molecules and a subset of 75 molecules, ET-Flow achieves performance comparable to Torsional Diffusion and MCF-S, but falls short of matching the performance of MCF-B and MCF-L. It's worth noting that MCF-B and MCF-L are significantly larger models, potentially affording them an advantage in generalization tasks. As part of our future work, we plan to scale up our model and conduct further tests to explore its performance in this regard.

*Table 8.* Generalization results on GEOM-XL.

| | AMR-P ↓ | | AMR-R ↓ | | # mols |
| | mean | median | mean | median | |
|---|---|---|---|---|---|
| GeoDiff | 2.92 | 2.62 | 3.35 | 3.15 | - |
| GeoMol | 2.47 | 2.39 | 3.30 | 3.14 | - |
| Tor. Diff. | 2.05 | 1.86 | **2.94** | 2.78 | - |
| MCF - S | 2.22 | 1.97 | 3.17 | 2.81 | 102 |
| MCF - B | 2.01 | 1.70 | 3.03 | 2.64 | 102 |
| MCF - L | **1.97** | **1.60** | **2.94** | **2.43** | 102 |
| ET-Flow (ours) | 2.31 | 1.93 | 3.31 | 2.84 | 102 |
| Tor. Diff. | 1.93 | 1.86 | 2.84 | 2.71 | 77 |
| MCF - S | 2.02 | 1.87 | 2.9 | 2.69 | 77 |
| MCF - B | 1.71 | 1.61 | 2.69 | 2.44 | 77 |
| MCF - L | **1.64** | **1.51** | **2.57** | **2.26** | 77 |
| ET-Flow (ours) | 2.00 | 1.80 | 2.96 | 2.63 | 75 |

## D. Training and Sampling Algorithm

The following algorithms go over the pseudo-code for the training and sampling procedure. For each molecule, we use up to 30 conformations with the highest boltzmann weights as provided by CREST (Pracht et al., 2024) similar to that of (Jing et al., 2022)

---

**Algorithm 1** Training procedure

---

**Input:** molecules $[G_0, ..., G_N]$ each with true conformers $[C_{G,1}, ...C_{G,K_G}]$, the harmonic prior $\rho_0$, learning rate $\alpha$, number of epochs $N_e$, initialized vector field $v_\theta$

**Output:** trained flow matching model $v_\theta$

**for** $i \leftarrow 1$ to $N_e$ **do**

    **for** each $G$ $[G_0, ..., G_N]$ **do**

        Sample $t \sim \mathcal{U}[0, 1]$

        Sample $C_1$ from $[C_{G,1}, ...C_{G,K_G}]$

        Sample prior $C_0 \sim \rho_0(G)$

        Align $C_0$ to $C_1$ using RMSD alignment

        $C_0 \leftarrow \text{RMSDAlign}(C_0, C_1)$

        Sample $C_t \sim \mathcal{N}\left(tC_1 + (1-t)C_0, \sigma^2 t(1-t)\mathbf{I}\right)$

        Calculate $u_t \leftarrow C_1 - C_0 + \frac{1-2t}{2\sqrt{t(1-t)}}z, z \sim \mathcal{N}(0, \mathbf{I})$

        Compute loss $\mathcal{L} \leftarrow \|v_\theta(t, C_t) - u_t\|^2$

        Take gradient step $\theta \leftarrow \theta - \alpha\nabla_\theta \mathcal{L}$

    **end for**

**end for**

---

**Algorithm 2** Inference procedure

---

**Input:** molecular graph $G$, number conformers $K$, number of sampling steps $N$

**Output:** predicted conformers $[C_1, ...C_K]$

**for** $C$ **in** $[C_1, ...C_K]$ **do**

    Sample prior $\hat{C} \sim \rho_0(G)$

    **for** n $\leftarrow 0$ to $N - 1$ **do**

        Set $t \leftarrow \frac{n}{N}$

        Set $\Delta t \leftarrow \frac{1}{N}$

        Predict $\hat{v} = v_\theta(t, \hat{C})$

        Update $\hat{C} = \hat{C} + \hat{v}\Delta t$

    **end for**

**end for**

---

**Algorithm 3** Stochastic Sampler

---

**Input:** molecular graph $G$, number conformers $K$, number of sampling steps $N$, stochasticity level $churn$, stochastic sampling range $[t_{min}, t_{max}]$

**Output:** predicted conformers $[C_1, ...C_K]$

**for** $C$ **in** $[C_1, ...C_K]$ **do**

    sample prior $\hat{C} \sim \rho_0(G)$

    **for** n $\leftarrow$ 0 to $N - 1$ **do**

        Set $t \leftarrow \frac{n}{N}, \Delta t \leftarrow \frac{1}{N}, \gamma \leftarrow \frac{churn}{N}$

        **if** $t \in [t_{min}, t_{max}]$ **then**

            Sample $\epsilon \sim N(0, I)$

            $\Delta \hat{t} \leftarrow \gamma(1 - t), \hat{t} \leftarrow max(t - \Delta \hat{t}, 0)$

            $\hat{C} \leftarrow \hat{C} + \Delta \hat{t}\sqrt{t^2 - \hat{t}^2}\epsilon$

            Predict $\hat{v} = v_\theta(\hat{t}, \hat{C})$

            Set $\Delta t \leftarrow \Delta t + \Delta \hat{t}$

        **else**

            Predict $\hat{v} = v_\theta(t, \hat{C})$

        **end if**

        Update $\hat{C} = \hat{C} + \hat{v}\Delta t$

    **end for**

**end for**

---

# E. Visualizations

Figure 5 shows randomly selected examples of sampled conformers from ET-Flow for GEOM-DRUGS. The left column is the reference molecule from the ground truth, and the remaining columns are samples generated with 50 sampling steps. Figure 6 showcases the ability for ET-Flow to generate quality samples with fewer sampling steps.

Reference

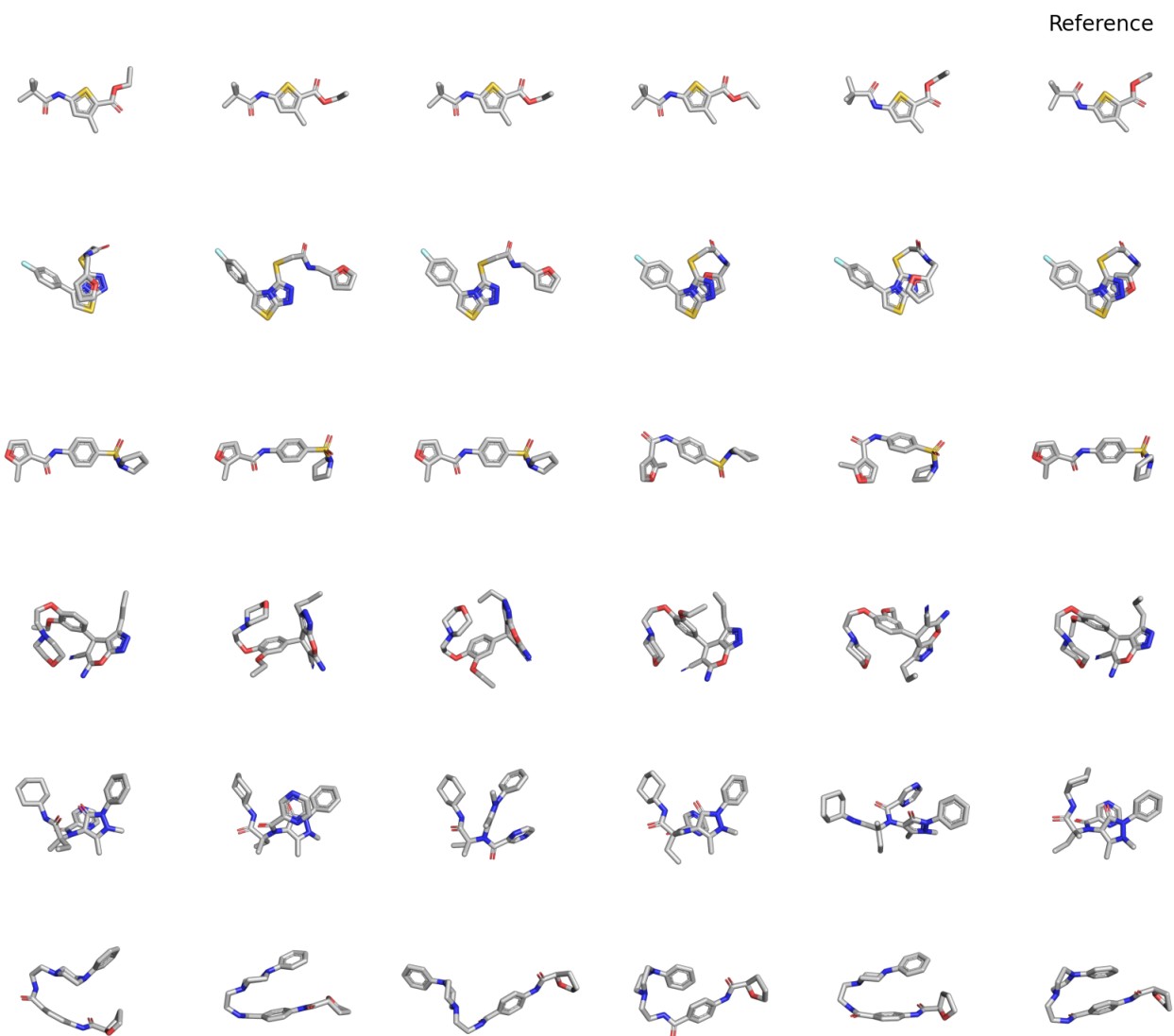

*Figure 5.* Examples of conformers of ground truth and ET-Flow for GEOM-DRUGS.

ET-Flow-5  ET-Flow-10  ET-Flow-20  ET-Flow-50  Reference

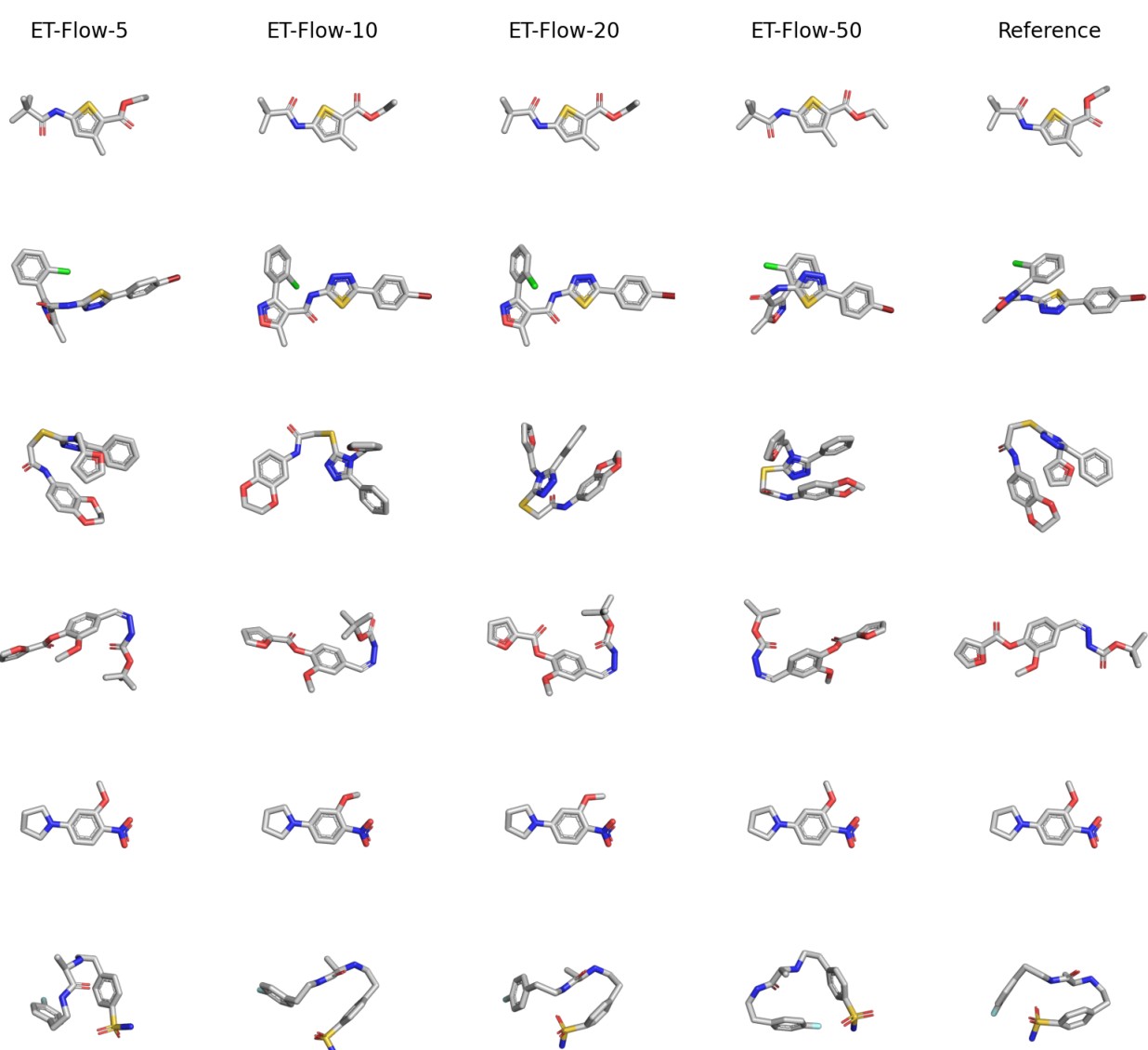

*Figure 6.* Examples of conformers of ground truth and ET-Flow for different number of sampling steps.