# OpenReview forum: "Equivariant Flow Matching for Molecular Conformer Generation"
_ICML.cc/2024/Workshop/ML4LMS — ML4LMS Poster_

### Official Review · Reviewer_FJNx · 2024-06-12
**Equivariant Flow Matching for Molecular Conformer Generation**

**Rating:** 6
**Confidence:** 3

**Review:**

Equivariant Flow Matching for Molecular Conformer Generation

---

### Official Review · Reviewer_1o3L · 2024-06-12
**ET-Flow is more efficient at generating molecular conformers, but methods of benchmarking improvement could be improved**

**Rating:** 6
**Confidence:** 4

**Review:**

**Summary**
The paper describes ET-Flow, a flow matching network that generates 3D molecule conformers with performance similar to existing models, albeit with a more efficient model.

**Strengths**
1. *Model Efficiency:* ET-Flow uses fewer parameters, a reduced number of sampling steps, and simpler model architecture ET-Flow is able to produce similar performance metrics (see next point) to other recent models.
2. *Quality of 3D Conformers:* the precision metrics for ET-Flow generated conformers show a modest improvement relative to other recent models (GeoDiff, GeoMol, Torsional Diff, MCF). For recall metrics, ET-Flow does not represent SOTA.

**Weaknesses**
1. *Model Efficiency:* the authors note that ET-Flow is able to produce an improvement (in some cases) over existing best-in-class models, albeit with a more efficient model. However, the inference time was not benchmarked. In virtual screens, multiple conformers must be generated for many molecules and this step is a well-known computational bottleneck. Ultimately, the impact / uptake of a model such as ET-Flow will depend on inference efficiency, so this should be examined.
2. *Out of Domain Performance:* existing benchmarks could also be improved by benchmarking on out-of-domain molecules since the datasets used are known to be somewhat limited in nature.

**Overall Recommendation**
A weak accept is recommended for "Equivariant Flow Matching for Molecular Conformer Generation". The authors endeavored to produce a more efficient model, which is a useful outcome, particularly when model inference is expected to be a computational bottleneck, as would be the case for a virtual screen (one common use for models such as ET-Flow). However, the paper could be significantly improved by addition of a benchmark of model inference times, which are highly relevant for such a bottleneck step.

---

### Official Review · Reviewer_DxJq · 2024-06-12
**Flow Matching for Conformer Generation**

**Rating:** 6
**Confidence:** 5

**Review:**

Thanks for submitting this. While the work is impressive, it doesn't seem to compare to current state-of-the-art methods.

I'd consider comparing ETFlow to MDF (https://arxiv.org/abs/2305.15586) which seems to score better across the board for recall + precision evaluation metrics.